# Weak Form Generalized Hamiltonian Learning

**Kevin L. Course**
University of Toronto
kevin.course@mail.utoronto.ca

**Trefor W. Evans**
University of Toronto
trefor.evans@mail.utoronto.ca

**Prasanth B. Nair**
University of Toronto
pbn@utias.utoronto.ca

## Abstract

We present a method for learning generalized Hamiltonian decompositions of ordinary differential equations given a set of noisy time series measurements. Our method simultaneously learns a continuous time model and a scalar energy function for a general dynamical system. Learning predictive models in this form allows one to place strong, high-level, physics inspired priors onto the form of the learnt governing equations for general dynamical systems. Moreover, having shown how our method extends and unifies some previous work in deep learning with physics inspired priors, we present a novel method for learning continuous time models from the weak form of the governing equations which is less computationally taxing than standard adjoint methods.

## 1 Introduction

While the bulk of dynamical system modeling has been historically limited to autoregressive-style models [1] and discrete time system identification tools [2, 3], recent years have seen the development of a diverse set of tools for directly learning continuous time models for dynamical systems from data. This includes the development of a rich set of methods for learning symbolic [4–11] and black-box [12–21] approximations of continuous-time governing equations using basis function regression and neural networks, respectively.

In terms of using neural networks to model continuous time ordinary differential equations (ODEs), a significant subset of these methods have focused on endowing the approximation with physics inspired priors. Making use of such priors allows models in this class to exhibit desirable properties by construction, such as being strictly

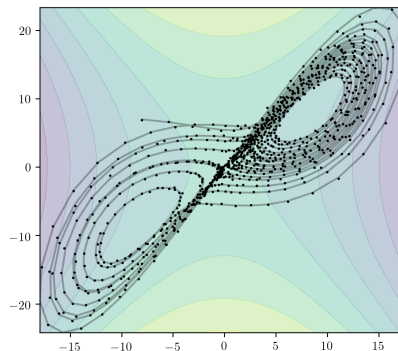

Figure 1: 2D slice of Lorenz '63 generalized Hamiltonian and trajectory

Hamiltonian [16, 17, 21] or globally stable [15]. While the existing literature presents a powerful suite of techniques for learning physics inspired parameterizations of ODEs, there remain limitations.

- Methods for leveraging physics inspired prior information on the form of the energy within the system are not applicable to general odd-dimensional ODEs.
- Methods for endowing ODEs with stability constraints require placing restrictions on the form of the Lyapunov function without directly placing a prior on the energy function. There are many systems for which we know a monotonically decreasing energy leads to stability.

- Methods for using neural networks to approximate continuous time ODEs require one to approximate the ODE derivatives or perform backpropagation by solving a computationally expensive adjoint ODE.

In this work, we address these issues by introducing a novel class of methods for learning generalized Hamiltonian decompositions of ODEs. Importantly, our method allows one to leverage high-level, physics inspired prior information on the form of the energy function even for odd-dimensional chaotic systems. This class of models generalizes previous work in the field by allowing for a broader class of prior information to be placed onto the energy function of a general dynamical system. Having introduced this new class of models, we present a weak form loss function for learning continuous time ODEs which is significantly less computationally expensive than adjoint methods while enabling more accurate learning than approximate derivative regression.

## 2 Generalized Hamiltonian Neural Networks

### 2.1 Generalized Hamiltonian Decompositions of Dynamical Systems

Our starting point is the generalized Hamiltonian decomposition proposed by Sarasola et al. [22] in the context of feedback synchronization of chaotic dynamical systems. In the present work we extend this decomposition from $\mathbb{R}^3$ to $\mathbb{R}^n$. To illustrate, consider an autonomous ODE of the form,

$$\dot{\mathbf{x}} = \mathbf{f}(\mathbf{x}), \tag{1}$$

where $\dot{(\cdot)}$ indicates temporal derivatives, $\mathbf{x} \in \mathbb{R}^n$, and $\mathbf{f} : \mathbb{R}^n \to \mathbb{R}^n$. The generalized Hamiltonian decomposition of the vector field, $\mathbf{f}$, is given by,

$$\mathbf{f}(\mathbf{x}) = (\mathbf{J}(\mathbf{x}) + \mathbf{R}(\mathbf{x})) \nabla H(\mathbf{x}), \tag{2}$$

where $\mathbf{J} : \mathbb{R}^n \to \mathbb{R}^{n \times n}$ is a skew-symmetric matrix, $\mathbf{R} : \mathbb{R}^n \to \mathbb{R}^{n \times n}$ is a symmetric matrix, and $H : \mathbb{R}^n \to \mathbb{R}$ is the generalized Hamiltonian energy function.

The generalized Hamiltonian decomposition in (2) is overly general; there are infinite choices for $\mathbf{J}$, $\mathbf{R}$, and $H$ which produce identical trajectories. We now show how the Helmholtz Hodge decomposition (HHD) can be used to impose constraints on the terms in (2) to ensure that the generalized Hamiltonian decomposition is physically meaningful.

Consider a HHD of the vector field in (2). The HHD extends the Helmholtz decomposition, which is valid in $\mathbb{R}^3$, to $\mathbb{R}^n$ [23][1]. For a vector field $\mathbf{f} : \mathbb{R}^n \to \mathbb{R}^n$, we make use of geometric algebra to define the HHD as,

$$\mathbf{f} = \mathbf{f}_1 + \mathbf{f}_2, \tag{3}$$

where $\nabla \cdot \mathbf{f}_1 = 0$, $\nabla \wedge \mathbf{f}_2 = \mathbf{0}$, and $\wedge$ is the geometric outer product[2]. This decomposes $\mathbf{f}$ into a sum of its divergence and curl-free components. The HHD suggests the imposition of the following divergence-free and curl-free constraints onto the decomposition in (2); they are $\nabla \cdot (\mathbf{J}\nabla H) = 0$ and $\nabla \wedge (\mathbf{R}\nabla H) = \mathbf{0}$ respectively. The following remarks discuss how these constraints naturally follow from considering a generalized Hamiltonian decomposition of a physical system.

**Remark 1** In physical systems governed by an autonomous ODE, energy variation occurs along with an associated change in phase space volume. Liouville's theorem states that the time derivative of a bounded volume in phase space for a vector field, $\mathbf{f}$, is given by $\dot{V}(t) = \int_{A(t)} (\nabla \cdot \mathbf{f}) d\mathbf{x}$, where $A(t)$ is a bounded set in phase space with volume $V(t)$ [22]. For an ODE decomposed as $\dot{\mathbf{x}} = \mathbf{f} = (\mathbf{J} + \mathbf{R}) \nabla H$, by requiring that $\mathbf{J}$ be skew-symmetric and $\nabla \cdot (\mathbf{J}\nabla H) = 0$ we see that,

$$\dot{H}(\mathbf{x}) = \nabla H(\mathbf{x})^T \dot{\mathbf{x}} = \nabla H(\mathbf{x})^T \mathbf{R}(\mathbf{x}) \nabla H(\mathbf{x}) \quad \& \quad \nabla \cdot \mathbf{f} = \nabla \cdot (\mathbf{R}\nabla H). \tag{4}$$

Noting that under this constraint the entire divergence is carried by $\mathbf{R}\nabla H$, we make use of the HHD to require that $\nabla \wedge (\mathbf{R}\nabla H) = \mathbf{0}$; this forces the entire curl onto $\mathbf{J}\nabla H$ without loss of generality. Hence energy variation occurs along with associated change in phase space volume and conserved dynamics are divergence free. In this way, a generalized Hamiltonian decomposition of an ODE which satisfies the divergence-free and curl-free constraints specified previously enforces that the generalized Hamiltonian, $H$, behaves similarly to a Hamiltonian for a real-world physical system.

**Remark 2** As expected, the generalized Hamiltonian decomposition reduces exactly to the standard Hamiltonian decomposition given further restrictions on the form of $\mathbf{J}$ and $\mathbf{R}$. We note that we can recover the standard Hamiltonian decomposition by setting,

$$\mathbf{J} = \begin{bmatrix} \mathbf{0} & \mathbf{1} \\ -\mathbf{1} & \mathbf{0} \end{bmatrix} \quad \& \quad \mathbf{R} = \mathbf{0}. \tag{5}$$

In this case the generalized Hamiltonian decomposition reduces *exactly* to the Hamiltonian decomposition as is used in related work by Bertalan et al. [16], Greydanus et al. [17], and Toth et al. [21].

## 2.2 Parameterizing Generalized Hamiltonian Decompositions

We have shown how decomposing an ODE in the form of a generalized Hamiltonian decomposition endows the ODE with a meaningful energy-like scalar function. The challenge then becomes how to parameterize the functions $\mathbf{J}$, $\mathbf{R}$, and $H$ such that the constraints of the decomposition are satisfied. In this work, we demonstrate how to parameterize these functions by neural networks such that the constraints are satisfied – we dub the resulting class of models *generalized Hamiltonian neural networks* (GHNNs). In the following exposition, $\mathcal{N} : \mathbb{R}^n \to \mathbb{R}$ will be used to refer to a neural network with a scalar valued output of the following form,

$$\mathcal{N}(\mathbf{x}) = \left( W_k \circ \sigma_{k-1} \circ W_{k-1} \circ \cdots \circ \sigma_1 \circ W_1 \right)(\mathbf{x}),$$

where $\circ$ indicates a composition of functions, $W_i$ indicates the application of an affine transformation, and each $\sigma_i$ indicates the application of a nonlinear activation function. Note that a unique solution to an initial value problem whose dynamics are defined by an autonomous ODE exists when $\mathbf{f}$ is Lipschitz continuous in $\mathbf{x}$ [25]. We use infinitely differentiable softplus activation functions unless otherwise noted due to differentiability requirements that will be discussed in the coming sections.

First we will discuss how to parameterize $\mathbf{J}$ such that the divergence-free constraint on the generalized Hamiltonian decomposition in (2) is satisfied by construction.

**Theorem 1.** *Let $\mathbf{J} : \mathbb{R}^n \to \mathbb{R}^{n \times n}$ be a skew-symmetric matrix whose $ij^{th}$ entry is given by $[\mathbf{J}]_{i,j} = g_{i,j}(\mathbf{x}_{\backslash ij})$, where $g_{i,j} = -g_{j,i} : \mathbb{R}^{n-2} \to \mathbb{R}$ is a differentiable function and $\mathbf{x}_{\backslash ij} = \{x_1, x_2, \ldots, x_n\} \backslash \{x_i, x_j\}$. Then it follows that,*

$$\nabla \cdot (\mathbf{J} \nabla H) = 0, \tag{6}$$

*where $H : \mathbb{R}^n \to \mathbb{R}$ is a twice differentiable function and $\backslash$ computes the difference between sets.*

The proof is given in Appendix A. In the present work we parameterize each $g_{i,j}$ by a neural network,

$$g_{i,j}(\mathbf{x}_{\backslash ij}) = \mathcal{N}(\mathbf{x}_{\backslash ij}). \tag{7}$$

Now we will develop some parameterizations for $\mathbf{R} \nabla H$ that will allow us to approximately satisfy the curl-free constraint on the decomposition in (2).

**Theorem 2.** *Let $V : \mathbb{R}^n \to \mathbb{R}$ and $H : \mathbb{R}^n \to \mathbb{R}$ be thrice and twice differentiable scalar fields respectively. If the Hessians of $V$ and $H$ are simultaneously diagonalizable, then it follows that,*

$$\nabla \wedge (\nabla^2 V \nabla H) = \mathbf{0}, \tag{8}$$

*where $\nabla^2$ denotes the Hessian operator.*

The proof is given in Appendix B. Unfortunately, parameterizing scalar functions $V$ and $H$ such that their Hessians are simultaneously diagonalizable requires that we compute the eigenvectors of $V$ or $H$ (see Appendix B.1 for such a parameterization). To avoid doing so, we consider two possible parameterizations for $\mathbf{R} \nabla H$. Let $\mathcal{N}_D : \mathbb{R}^n \to \mathbb{R}$ and $\mathcal{N}_v : \mathbb{R}^n \to \mathbb{R}$ be neural networks. The two parameterizations we consider are,

$$\mathbf{R} \nabla H = \nabla \mathcal{N}_D(\mathbf{x}) \quad \text{and} \quad \mathbf{R} \nabla H = \nabla^2 \mathcal{N}_v(\mathbf{x}) \nabla H(\mathbf{x}). \tag{9}$$

The first parameterization in (9) is curl-free by construction owing to the definition of the gradient operator. The second parameterization in (9) is not guaranteed to be curl-free but penalty methods can be used to enforce the constraint in practice. Note that the curl-free constraint is only intended to

limit the possible solution space for $H$ to make the energy function more meaningful. For this reason, exactly satisfying the constraint is not required.

While the first parameterization is cheaper to compute and it satisfies the curl-free constraint by construction, the second parameterization allows for a richer set of priors to be placed on the form of the generalized Hamiltonian. This is discussed in depth in Section 2.3. Finally, note that we are not required to explicitly compute $\nabla^2 \mathcal{N}_v$ when computing $\mathbf{R}\nabla H$. Instead, we only require the product between $\nabla^2 \mathcal{N}_v$ and $\nabla H$ when computing the ODE model output.

## 2.3 Choices for Priors on the Generalized Hamiltonian

We have presented a number of parameterizations for $\mathbf{J}$ and $\mathbf{R}$ in the previous section. This section demonstrates the power of the generalized Hamiltonian formalism by explaining how these different parameterizations can be mixed and matched to leverage different priors on the form of the governing equations. Unless otherwise noted, we will use the same parameterization for $\mathbf{J}$ in all cases.

**Globally Asymptotically Stable & Energy Decaying**   By globally asymptotically stable we mean systems which always converge to $\mathbf{x} = \mathbf{0}$ in finite time; one example of such a system is a pendulum with friction. To enforce global stability, we choose the second parameterization in (9) for $\mathbf{R}\nabla H$ so that $\mathbf{R}\nabla H = \nabla^2 \mathcal{N}_v(\mathbf{x})\nabla\mathcal{N}_H(\mathbf{x})$, where $\mathcal{N}_v$ is chosen to be an input concave neural network [26]. Furthermore we set $H$ as follows,

$$H = \mathcal{N}_H(\mathbf{x}) = ReHU(\mathcal{N}(\mathbf{x}) - \mathcal{N}(\mathbf{0})) + \epsilon \mathbf{x}^T \mathbf{x}, \tag{10}$$

where $ReHU$ is the rectified Huber unit as described by Kolter and Manek [15]. Since $\mathcal{N}_v$ is concave, its Hessian is negative definite and we see that the energy variation along trajectories of the system must be strictly decreasing, i.e. $\dot{H} = \nabla H^T \mathbf{R}\nabla H = \nabla\mathcal{N}_H(\mathbf{x})^T \nabla^2\mathcal{N}_v(\mathbf{x})\nabla\mathcal{N}_H(\mathbf{x}) < 0$.

In addition, due to our parameterization for $\mathcal{N}_H$, we see that $H(\mathbf{x}) > 0 \,\forall \mathbf{x} \neq \mathbf{0}$, $H(\mathbf{0}) = 0$, and $H(\mathbf{x}) \to \infty$ as $\mathbf{x} \to \infty$. We see that $\mathcal{N}_H$ then acts as a globally stabilizing Lyapunov function for the system [27]. Note that even for a random initialization of the weights in our model the ODE will be strictly globally stable at $\mathbf{x} = \mathbf{0}$. Furthermore, unlike the work of Kolter and Manek [15], (i) we are not required to place convexity restrictions onto the form of our Lyapunov function and (ii) we are placing a prior directly onto the energy function of the state rather than an arbitrary scalar function.

**Locally Asymptotically Stable & Energy Decaying**   By locally asymptotically stable we mean systems for which we know energy strictly decreases along trajectories of the system and there are multiple energy configurations which the system could converge to (ie. there are potentially multiple regions of local stability). One such example of a system is a particle in a double potential well with energy decay. As before, to enforce this prior we choose the second parameterization in (9) for $\mathbf{R}\nabla H$ so that $\mathbf{R}\nabla H = \nabla^2 \mathcal{N}_v(\mathbf{x})\nabla\mathcal{N}_H(\mathbf{x})$ where $\mathcal{N}_v(\mathbf{x})$ is chosen to be an input concave neural network [26]. Furthermore, we parameterize $H$ as,

$$H = \mathcal{N}_H(\mathbf{x}) = \sigma\left(\mathcal{N}(\mathbf{x})\right) - \sigma\left(\mathcal{N}(\mathbf{0})\right) + \epsilon \mathbf{x}^T \mathbf{x}. \tag{11}$$

This parameterization enforces the condition that $\dot{H} < 0$ along trajectories of the system and that $\mathcal{N}_H(\mathbf{x}) + \mathcal{N}_H(\mathbf{0}) > 0$, $\mathcal{N}_H(\mathbf{0}) = \mathbf{0}$, and $\mathcal{N}_H(\mathbf{x}) \to \infty$ as $\mathbf{x} \to \infty$. This ensures that the trajectory will stabilize to some fixed point even for a randomly initialized set of weights.

**Generalized Hamiltonian is Conserved**   We can also enforce that the generalized Hamiltonian be conserved along trajectories of the system by construction. To do so, we can choose any parameterization for $H$ and set $\mathbf{R} = \mathbf{0}$. In this case we see that $\dot{H} = 0$ along trajectories of the system by construction. Note that we have not needed to assume that our system is Hamiltonian or that our system can be described in terms of a Lagrangian. Our approach is valid even for odd-dimensional systems meaning that it is applicable even to surrogate models of complex systems which need not be derived from the laws of dynamics.

**Setting Energy Flux Rate**   In addition to the strong priors on the form of the energy function listed above, it is also possible to place soft priors on the form of the energy function. For example, we can regularize the loss function with some known energy transfer rate. Consider weather modeling for example; while placing strong forms of prior information onto the form of the energy function may be

challenging, it may be possible to estimate the energy flux rate for some local climate given the time of year, latitude, etc. For example, given some nominal energy flux rate measured at $m$ time instants, $\{\dot{H}_{\text{nom}}(t_i)\}_{i=1}^m$, an arbitrary parameterization of $H$ given by $\mathcal{N}_H$, and an arbitrary parameterization of $\mathbf{R}\nabla H$ given by $\nabla\mathcal{N}_D$ we can add $\frac{1}{m}\sum_{i=1}^m ||\nabla\mathcal{N}_H(\mathbf{x}(t_i))^T\nabla\mathcal{N}_D(\mathbf{x}(t_i)) - \dot{H}_{\text{nom}}(t_i)||_2^2$ to the loss function at training time. As will be demonstrated by the numerical studies in Section 4, such regularization can help heal identifiability issues with the generalized Hamiltonian decomposition when other forms of prior information are not available.

**Known Generalized Hamiltonian**   This is a useful prior as it is often straightforward to identify the total energy of a system without being able to write-down all sources of energy addition or depletion. To this end, we consider an extremely flexible parameterization for the generalized Hamiltonian,

$$\mathbf{f} = \mathbf{W}(\mathbf{x})\nabla H(\mathbf{x}), \tag{12}$$

where $\mathbf{W} : \mathbb{R}^n \to \mathbb{R}^{n\times n}$ is a square matrix and $H$ is the known energy function. From $\mathbf{W}$ we can easily recover, $\mathbf{J} = (\mathbf{W} - \mathbf{W}^T)/2$ and $\mathbf{R} = (\mathbf{W} + \mathbf{W}^T)/2$. The study in Section 4.3 provides an example of the interpretability gained by learning a decomposition of an ODE in this form.

## 3  Parameter Estimation for GHNNs

We now consider the problem of efficiently estimating the parameters of GHNNs given a set of noisy time series measurements. After a brief review of common methods for parameter estimation, we propose a novel procedure for learning from the weak form of the governing equations. To the best of the knowledge of the authors, this method has not been proposed in the context of deep learning. This method drastically reduces the computational cost of learning continuous time models as compared to adjoint methods while being significantly more robust to noise than derivative regression.

We make use of the notation $\dot{\mathbf{x}} = \mathbf{f}_\theta(\mathbf{x})$ to indicate a parameterized ODE. We collect $m$ trajectories of length $T$ of the state, $\mathbf{x}$. We will use the short hand notation $\mathbf{x}_j^{(i)}$ to indicate the measurement of the state at time instant $t_j$, for trajectory $i$. Our dataset is then as follows: $\mathcal{D} = \{\mathbf{x}_1^{(i)}, \mathbf{x}_2^{(i)}, \ldots, \mathbf{x}_T^{(i)}\}_{i=1}^m = \{\mathbf{X}^{(i)}\}_{i=1}^m$, where $\mathbf{X}^{(i)}$ indicates the collection of state measurements for the $i^{th}$ trajectory.

**Review of Methods**   In maximum likelihood state regression methods, the parameters are estimated by solving the optimization problem,

$$\theta^* = \arg\max_\theta \frac{1}{m}\sum_{i=1}^m \log p_\theta(\mathbf{x}_2^{(i)}, \mathbf{x}_3^{(i)}, \ldots, \mathbf{x}_T^{(i)}|\mathbf{x}_1^{(i)}),$$
$$\text{subject to:}\quad \dot{\mathbf{x}}_j^{(i)} = \mathbf{f}_\theta(\mathbf{x}_j^{(i)})\quad \forall j \in [1, 2, ..., T], \forall i \in [1, 2, ..., m]. \tag{13}$$

In other words, we integrate an initial condition, $\mathbf{x}_1^{(i)}$, forward using an ODE solver and maximize the likelihood of these forward time predictions given measurements of the state – hence we refer to this class of methods as "state regression". This optimization problem can be iteratively solved using adjoint methods with a memory cost that is independent of trajectory length [14]. While the memory cost of these methods are reasonable, a limitation of these methods is that they are computationally expensive. For example, the common Runge-Kutta 4(5) adaptive solver requires a minimum of six evaluations of the ODE for each time step [14].

Derivative regression techniques attempt to reduce the computational cost of state regression by performing regression on the derivatives directly. While in some circumstances derivatives of the state can be measured directly, most often these derivatives must first be estimated at each time instant using finite difference schemes [28]. This yields the augmented dataset, $\tilde{\mathcal{D}} = \{(\mathbf{x}_1^{(i)}, \dot{\mathbf{x}}_1^{(i)}), (\mathbf{x}_2^{(i)}, \dot{\mathbf{x}}_2^{(i)}), \ldots (\mathbf{x}_T^{(i)}, \dot{\mathbf{x}}_T^{(i)})\}_{i=1}^m$. In maximum likelihood derivative regression, the optimal ODE parameters are estimated as,

$$\theta^* = \arg\max_\theta \frac{1}{mT}\sum_{i=1}^m\sum_{j=1}^T \log p_\theta(\dot{\mathbf{x}}_j^{(i)}|\mathbf{x}_j^{(i)}). \tag{14}$$

While this method is less computationally taxing than state regression as it does not require an expensive ODE solver, it is limited by the fact that derivative estimation is highly inaccurate in the presence of even moderate noise [8].

**Weak form learning of GHNNs** While derivative regression [5, 15–17, 20] and state regression [14, 18, 21] are well-known in the deep learning literature, learning ODEs from the weak form of the governing equations has only been used in the context of sparse basis function regression as far as the authors are aware [8, 29].

In the present work we show how to use the weak form of the governing equations in the context of learning deep models for ODEs. This method allows one to drop the requirement of estimating the state derivatives at each time step without having to backpropogate through an ODE solver or solving an adjoint ODE – drastically cutting the computational cost of learning deep continuous time ODEs. Pantazis and Tsamardinos [8] and Schaeffer and McCalla [29] independently showed how the idea of working with the weak form of the governing equations could be used in the context of sparse regression to learn continuous time governing equations using data corrupted by significantly more noise than is possible with derivative regression.

To derive the weak form loss function, we multiply the parameterized ODE by a time dependent sufficiently smooth[3] continuous test function $v : \mathbb{R} \to \mathbb{R}$, integrate over the time window of observations

$$\int_{t_1}^{t_T} v\dot{\mathbf{x}} dt = \int_{t_1}^{t_T} v\mathbf{f}_\theta(\mathbf{x}) dt, \tag{15}$$

and integrate by parts,

$$v\mathbf{x}\Big|_{t_1}^{t_T} - \int_{t_1}^{t_T} \dot{v}\mathbf{x} dt = \int_{t_1}^{t_T} v\mathbf{f}_\theta(\mathbf{x}) dt. \tag{16}$$

In order to reduce this infinite dimensional problem into a finite set of equations, we introduce a dictionary of $K$ test functions $\{\psi_1(t), \psi_2(t), \ldots, \psi_K(t)\}$. This Petrov-Galerkin discretization step leads to,

$$\psi_k\mathbf{x}\Big|_{t_1}^{t_T} - \int_{t_1}^{t_T} \dot{\psi_k}\mathbf{x} dt = \int_{t_1}^{t_T} \psi_k\mathbf{f}_\theta(\mathbf{x}) dt \quad \forall k \in [1, 2, \ldots, K]. \tag{17}$$

Assuming the time measurements are sufficiently close together, we can efficiently estimate the integrals in (17) using standard quadrature techniques. The weak form of the governing equations leads to a new maximum likelihood objective,

$$\theta^* = \arg\max_\theta \frac{1}{m} \sum_{i=1}^{m} \sum_{k=1}^{K} \log p_\theta \left( \psi_k\mathbf{x}^{(i)}\Big|_{t_1}^{t_T} - \int_{t_1}^{t_T} \dot{\psi_k}\mathbf{x}^{(i)} dt \Big| \mathbf{x}^{(i)} \right). \tag{18}$$

This weak derivative regression method allows us to eliminate the requirement of estimating derivatives or performing the expensive operations of differentiating through an ODE solver or solving an adjoint ODE.

# 4 Numerical Studies[4]

We compare our approach (GHNN) to a fully connected neural network (FCNN) and Hamiltonian neural network (HNN). All models were trained on an Nvidia GeForce GTX 980 Ti GPU. We used PyTorch [30] to build our models, Chen et al.'s [14] "torchdiffeq" in experiments that used state regression, and the Huber activation function from Kolter and Manek [15]. Unless otherwise noted, we will use the default settings for the adjoint ODE solvers offered in Chen et al.'s package; at the time of writing, this includes a relative tolerance of $10^{-6}$ and an absolute tolerance of $10^{-12}$ with a Runge-Kutta(4)5 adaptive ODE solver. The metrics used in the coming sections are described in detail in Appendix D. A description of all the architectures of the neural networks used in this work can be found in Appendix E.

For all experiments that use weak derivative regression, the test space is spanned by 200 evenly spaced Gaussian radial basis functions with a shape parameter of 10 over each mini-batch integration window; this is explained in more detail in Appendix H. A description of mini-batching hyperparameters specific to learning ODEs can be found in Appendix G.

### 4.1 Comparison of Methods for Learning ODE Models

We will attempt to learn an approximation to a nonlinear pendulum using a FCNN with a weak derivative loss function, a derivative regression loss function, and a state regression loss function. We collect measurements of the pendulum state corrupted by Gaussian noise with a standard deviation of $0.1$ as it oscillates towards its globally stable equilibrium along two independent trajectories sampled at a frequency of 50Hz for 20 seconds.

The error in the states, its derivatives, and training time for the three methods of parameter estimation are given in Table 1. Note that at this level of noise, derivative regression learnt an ODE model which diverged in finite time and hence the prediction error could not be calculated.

Table 1: Comparison of our approach (weak form regression) to state regression and derivative regression for learning a continuous-time model of a nonlinear pendulum

| Metric | Approach | | |
| --- | --- | --- | --- |
| | Weak form regression | Derivative regression | State regression |
| State Error | $\mathbf{0.17 \pm 0.05}$ | Diverged | $0.48 \pm 0.24$ |
| Derivative Error | $\mathbf{0.15 \pm 0.08}$ | $1.35 \pm 0.76$ | $0.38 \pm 0.13$ |
| Train Time | 34s | $\mathbf{29}$s | 25min 46s |

We see that weak derivative matching has comparable performance to state regression while requiring substantially less run-time. A more extensive study, which includes a variety of measurement sampling frequencies, led to similar trends (see Appendix I). In the studies that follow we shall therefore exclusively focus on weak derivative regression.

### 4.2 Example Problems

**Nonlinear Pendulum** The generalized Hamiltonian decomposition for a damped nonlinear pendulum is provided in Appendix C. The experiment setup is the same as in Appendix E.

We make the assumption that the system is asymptotically globally stable at $\mathbf{x} = \mathbf{0}$ as we only concern ourselves with initial conditions sufficiently close to the origin. Under these assumptions, we place a globally stable prior onto the form of the generalized Hamiltonian energy function. Recall that even for a randomly initialized set of weights, the ODE model is guaranteed to stabilize to $\mathbf{x} = \mathbf{0}$. Note that unlike existing methods in the literature, we are able to place a globally stabilizing prior onto our model structure while simultaneously learning the underlying generalized Hamiltonian.

In Figure 2 we observe that the trajectories produced by the learnt model align well with trajectories produced by the true underlying equations and the generalized Hamiltonian energy function. Furthermore, in Figure 4 we observe that our model learnt the important qualitative features of the vector field and generalized Hamiltonian. The performance of GHNNs on this problem is compared to FCNNs and HNNs in Table 2.

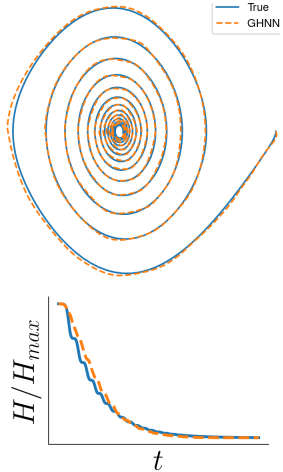

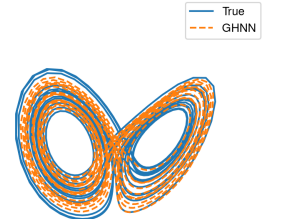

Figure 2: GHNN predicted pendulum trajectory

**Lorenz '63 System** A generalized Hamiltonian decomposition of the governing equations for the Lorenz system can be found in Appendix C. We collect measurements of the state corrupted by Gaussian noise with a standard deviation of 0.1 for 20 seconds at a sampling frequency of 250Hz along 21 independent initial conditions.

Note that without prior information, this decomposition is not unique; in other words, there are multiple generalized Hamiltonian energy functions which would well-represent the dynamics. As before, we collect noisy measurements of the system state and attempt to learn the dynamics in the form of a generalized Hamiltonian decomposition.

Figure 3: Lorenz predicted trajectory example

In this experiment we place a soft energy flux rate prior on the form of the generalized Hamiltonian energy function. We see that the model is able to capture the fact the system decays to a strange attractor as is shown in Figure 3. Note that because the governing equations are chaotic, we expect to only be able to capture qualitative aspects of the trajectory. Furthermore, we see in Figure 5 that we were able to learn the generalized Hamiltonian and vector field. It should be noted that without this soft prior, we would would not be able to learn the true generalized Hamiltonian; this is discussed in Appendix K. The performance of GHNNs on this problem is compared to FCNNs in Table 2. Note that HNNs are not applicable to this problem as the state space is odd-dimensional.

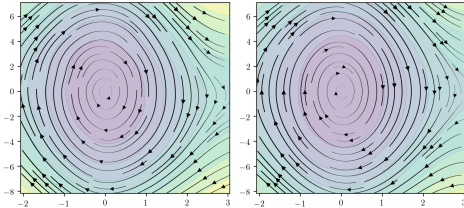

Figure 4: Learnt (L) and true (R) pendulum generalized Hamiltonian and vector field

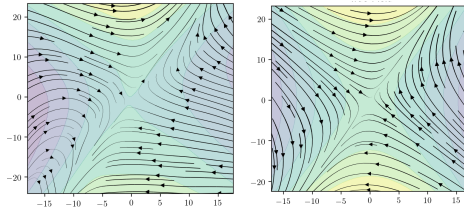

Figure 5: Learnt (L) and true (R) Lorenz '63 generalized Hamiltonian and vector field

**Benchmarking Summary**   We have applied our approach to three problems in this subsection: the nonlinear pendulum, the Lorenz '63 system, and the Duffing oscillator (see Appendix J). Typically a HNN would not be applied to these systems as they are not energy conserving, but we do so here to demonstrate the strength of our method when this knowledge is leveraged. We use the same metrics as are defined in Appendix D. We do not compute the state error for the Lorenz '63 system because the governing equations are chaotic. In all cases, GHNNs perform approximately as well as the flexible FCNN models while simultaneously learning the generalized Hamiltonian energy function and the energy cycle for the system.

GHNNs allow us to pursue "what if" scenarios related to the form of the energy function such as: what if the rate of energy transfer is halved or what if the mechanism of energy transfer is altered? To the best of the author's knowledge, this work is the first to demonstrate this ability in the context of general odd dimensional ODEs with a broad class of possible priors.

Table 2: Comparison of our approach (GHNNs) to FCNNs and HNNs

| Model | N.L. Pendulum | | Duffing | | Lorenz '63 |
| | State Error | Derivative Error | State | Derivative | Derivative |
| --- | --- | --- | --- | --- | --- |
| GHNN w/ prior | $0.08 \pm 0.10$ | $0.07 \pm 0.21$ | $0.31 \pm 0.68$ | $0.06 \pm 0.02$ | $5.24 \pm 32.78$ |
| FCNN | $0.04 \pm 0.03$ | $0.43 \pm 0.05$ | $0.26 \pm 0.63$ | $0.03 \pm 0.01$ | $3.46 \pm 14.02$ |
| HNN | $3.20 \pm 2.27$ | $0.08 \pm 0.10$ | $1.61 \pm 0.88$ | $0.12 \pm 0.08$ | N/A |

## 4.3  Discovering Energy Sources & Losses

To demonstrate the power of the interpretability afforded by the generalized Hamiltonian approach, we consider the problem of discovering where energy sources and losses occur in a dynamical system. We consider learning the dynamics of an $N$-body problem in two-dimensions where the particles are subjected to a non-conservative force field (i.e. with a non-vanishing curl) such that the energy of the system is not constant. It is assumed that the energy function $H$ is known *a priori*, and we therefore choose the parameterization (12) where we learn $\mathbf{W}(\mathbf{x}) = \mathbf{J}(\mathbf{x}) + \mathbf{R}(\mathbf{x})$ together as the output of an unconstrained neural network. A dataset was generated by integrating the dynamics forward in time using $N = 12$ particles, giving $n = 4N = 48$ state variables. Further details of the force field, governing dynamics and experimental setup are provided in Appendix F.

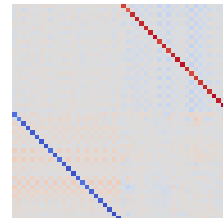

Figure 6: Learnt $\mathbf{J}$ for $N$-body problem

After training, we can recover $\mathbf{J}$ and $\mathbf{R}$. Matrix $\mathbf{J}$ is shown in Figure 6 where the discovered structure closely resembles (5) as expected since the (conservative part) of the $N$-body dynamics is Hamiltonian.

We now use the matrix $\mathbf{R}$ to discover which state variables are contributing to energy loss or gain. Specifically, $\nabla H \cdot \mathbf{R}\nabla H$ gives the instantaneous energy flux contributed by each state variable at a point $\mathbf{x}$, where $\cdot$ is the element-wise product. We visualize this energy flux breakdown in Figure 7, where colour indicates the flux contributed by each particle (summing the contribution from velocity and position variables). The location, $\mathbf{x}$, in state space is also shown through the particle positions and velocities, which are given by the black arrows. The grey arrows show the magnitude and direction of the non-conservative force field. As expected, positive energy flux is observed

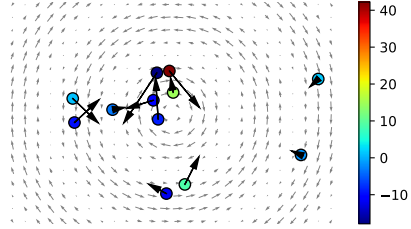

Figure 7: Instantaneous energy flux per particle for an $N$-body problem in a non-conservative force field. $\dot{H} = 3.98$

when the force vector and velocity vector are aligned since the force field would contribute to a particle's kinetic energy. Conversely, a negative energy flux is observed when the force vector and velocity vector are in opposite directions. For example, see the dark red and blue particles in the upper-center of Figure 7. Such an analysis could be helpful to discover and diagnose energy sources and losses in real dynamical systems.

## 5   Related Works

**Learning Hamiltonians**   Bertalan et al. [16], Greydanus et al. [17], and Toth et al. [21] independently developed methods for learning a Hamiltonian decomposition of an ODE. Hamiltonian systems can be roughly defined as even dimensional systems (ie. $\mathbf{x} \in \mathbb{R}^{2n}$) which are energy conserving. More recently, Zhong et al. [31] extended this work for learning ODEs governing Hamiltonian systems with control and energy dissipation [32]. We have shown in Section 2.3 how the generalized Hamiltonian formalism reduces exactly to the Hamiltonian formalism when further restrictions are placed onto the form of $\mathbf{J}$ and $\mathbf{R}$. Importantly, the generalized Hamiltonian formalism is applicable to even odd-dimensional chaotic systems with energy transfer.

**Learning Lagrangians**   Lutter et al. [19] showed how to learn Lagrangians for systems where the kinetic energy is an inner product of the velocity. By learning Lagrangians rather than Hamiltonians they could learn physically meaningful dynamics when only measurements of state in non-canonical coordinates were available. Their formulation requires measuring the generalized forces in addition to the system state. Cranmer et al. [20] later expanded on this work to systems where the kinetic energy was no longer an inner product of the velocity however they only considered conservative systems in their formulation. Note that like the Hamiltonian formalism, the Lagrangian formalism implies a state space which is even dimensional. Again, a key distinction with the present work is that the generalized Hamiltonian formalism does not require the state space to be even-dimensional or that we necessarily know the source of energy addition or depletion.

**Learning Stable Dynamics**   Kolter and Manek [15] presented a method for learning an ODE which is globally asymptotically stable by construction. They enforced global asymptotic stability by simultaneously learning a model for an ODE and a Lyapunov function with a single global minimum and no local minima. In the present work, we have shown how to place a broader set of priors directly onto the form of the energy function of the system – rather than an arbitrary scalar function of the state – without having to place convexity restrictions onto the form of the generalized Hamiltonian.

## 6   Conclusion

This paper made two main contributions. The first contribution shows how to learn a generalized Hamiltonian decomposition of an ODE. This decomposition simultaneously learns a generalized Hamiltonian energy function and a black-box ODE model; learning ODEs in this form allows us to place strong, high-level, physics inspired priors onto the form of the energy within the system. Importantly, this decomposition is valid for a broad class of ODEs including odd-dimensional, nonconservative systems. The second contribution of this work is in demonstrating how to learn deep continuous time models of ODEs from the weak form of the governing equations. We have shown how learning continuous time models using this formulation is significantly faster than using adjoint methods while simultaneously being more robust to noise than derivative matching methods.

## Broader Impact

Since this work is in large part theoretical in nature there are few ethical considerations directly related to this work. In terms of broader impact, this work builds on a long line of work which seeks to build better models for dynamical systems. The long term intent of work in this field is to learn better models of real world systems which currently evade first-principles-based modeling; for example, this has potential applications in climate science, financial markets, and disease outbreak modeling. In addition, this work specifically has presented a novel method for placing strong, high-level, physics informed priors onto the form of the equations governing nonlinear dynamical system to directly learn an ODE and a generalized Hamiltonian from noisy measurements of the system state. We hope this work inspires further development on learning physics inspired parameterizations of dynamical systems.

## Acknowledgements

This research is funded by grants from NSERC and the Canada Research Chairs program.

## Footnotes

[1]We consider a limited version of the HHD for decomposing vector fields specifically.

[2]See Macdonald [24] for more details on geometric algebra and calculus.

[3]In our numerical studies we use $C^\infty$ test functions.

[4]Code can be found online at: https://github.com/coursekevin/weakformghnn.

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
