[Supplementary Material]

## A  Proof of Divergence Free Parameterization

**Theorem 1.** *Let* $\mathbf{J} : \mathbb{R}^n \to \mathbb{R}^{n \times n}$ *be a skew-symmetric matrix whose* $ij^{th}$ *entry is given by* $[\mathbf{J}]_{i,j} = g_{i,j}(\mathbf{x}_{\setminus ij})$, *where* $g_{i,j} = -g_{j,i} : \mathbb{R}^{n-2} \to \mathbb{R}$ *is a differentiable function and* $\mathbf{x}_{\setminus ij} = \{x_1, x_2, \ldots, x_n\} \setminus \{x_i, x_j\}$. *Then it follows that,*

$$\nabla \cdot (\mathbf{J} \nabla H) = \mathbf{0}, \tag{19}$$

*where* $H : \mathbb{R}^n \to \mathbb{R}$ *is a twice differentiable function and* $\setminus$ *computes the difference between sets.*

*Proof.* The proof follows from evaluating the divergence of the parameterization. Noting that $\mathbf{J}$ is skew-symmetric (hence $[\mathbf{J}]_{i,j} = -[\mathbf{J}]_{j,i}$ and $[\mathbf{J}]_{i,i} = 0$), we can write the divergence of $\mathbf{J}\nabla H$ as,

$$
\begin{aligned}
\nabla \cdot (\mathbf{J} \nabla H) &= \sum_{i=1}^{N} \partial_{x_i} \sum_{j=1}^{N} [\mathbf{J}]_{i,j} \, \partial_{x_j} H, \\
&= \sum_{i<j}^{N} \sum_{j=1}^{N} \partial_{x_i} [\mathbf{J}]_{i,j} \, \partial_{x_j} H + \partial_{x_j} [\mathbf{J}]_{j,i} \, \partial_{x_i} H, \\
&= \sum_{i<j}^{N} \sum_{j=1}^{N} \partial_{x_i} [\mathbf{J}]_{i,j} \, \partial_{x_j} H - \partial_{x_j} [\mathbf{J}]_{i,j} \, \partial_{x_i} H, \\
&= \sum_{i<j}^{N} \sum_{j=1}^{N} [\mathbf{J}]_{i,j} \left( \partial_{x_i, x_j} H - \partial_{x_i, x_j} H \right) + \partial_{x_i} [\mathbf{J}]_{i,j} \, \partial_{x_j} H - \partial_{x_j} [\mathbf{J}]_{i,j} \, \partial_{x_i} H,
\end{aligned}
$$

where $\partial_{x_i}$ indicates a partial derivative with respect to $x_i$. We see that $(\partial_{x_i, x_j} H - \partial_{x_i, x_j} H) = 0$ for any twice differentiable $H$ and that $\partial_{x_i} [\mathbf{J}]_{i,j} = \partial_{x_j} [\mathbf{J}]_{i,j} = 0$ when $[\mathbf{J}]_{i,j} = g_{i,j}(\mathbf{x}_{\setminus ij})$. $\qquad\square$

## B  Proof of Curl Free Parameterization

**Theorem 2.** *Let* $V : \mathbb{R}^n \to \mathbb{R}$ *and* $H : \mathbb{R}^n \to \mathbb{R}$ *be thrice and twice differentiable scalar fields respectively. If the Hessians of* $V$ *and* $H$ *are simultaneously diagonalizable, then it follows that,*

$$\nabla \wedge (\nabla^2 V \nabla H) = \mathbf{0}, \tag{20}$$

*where* $\nabla^2$ *denotes the Hessian operator.*

*Proof.* The proof follows from evaluating the curl of the the expression.

$$
\begin{aligned}
\nabla \wedge \nabla^2 V \nabla H &= \partial_{x_i} \sum_k \partial_{x_j, x_k} V \partial_{x_k} H - \partial_{x_j} \sum_k \partial_{x_i, x_k} V \partial_{x_k} H, \quad \forall i < j, \\
&= \sum_k \left( \partial_{x_i} \partial_{x_j, x_k} V - \partial_{x_j} \partial_{x_i, x_k} V \right) \partial_{x_k} H + \\
&\qquad \left( \partial_{x_j, x_k} V \partial_{x_k, x_i} H - \partial_{x_i, x_k} V \partial_{x_k, x_j} H \right) \quad \forall i < j.
\end{aligned}
\tag{21}
$$

We see that $\left( \partial_{x_i} \partial_{x_j, x_k} V - \partial_{x_j} \partial_{x_i, x_k} V \right) = 0$ by construction. Furthermore, we can write,

$$\sum_k \left( \partial_{x_j, x_k} V \partial_{x_k, x_i} H - \partial_{x_i, x_k} V \partial_{x_k, x_j} H \right) = \left[ \nabla^2 V \nabla^2 H - \nabla^2 H \nabla^2 V \right]_{LT}, \tag{22}$$

where $[\mathbf{A}]_{LT}$ extracts the lower triangular elements from $\mathbf{A} \in \mathbb{R}^{n \times n}$. We see that (22) will be zero by construction if $\nabla^2 V$ and $\nabla^2 H$ commute.

$\nabla^2 V$ and $\nabla^2 H$ are symmetric real matrices hence they are always diagonalizable. Because they are simultaneously diagonalizable (ie. they share a common set of eigenvectors [33]), $\left[ \nabla^2 V \nabla^2 H - \nabla^2 H \nabla^2 V \right]_{LT} = \mathbf{0} \implies \nabla \wedge (\nabla^2 V \nabla H) = \mathbf{0}$. $\qquad\square$

## B.1  Spectral Curl-Free Parameterization

In this section we discuss a spectral parameterization for $\mathbf{R}$ which is curl free by construction. Note that we have not implemented this parameterization due to computational limitations associated with computing the eigenvectors of Hessians.

Let the paramterization of $H = \mathcal{N}_H$ with Hessian matrix $\nabla^2 \mathcal{N}_H(\mathbf{x})$ be diagonalizable as follows

$$\nabla^2 \mathcal{N}_H(\mathbf{x}) = \mathbf{Q}(\mathbf{x})\mathbf{T}(\mathbf{x})\mathbf{Q}(\mathbf{x})^{-1}, \tag{23}$$

where $\mathbf{Q}(\mathbf{x}) : \mathbb{R}^n \to \mathbb{R}^{n \times n}$ is a unitary matrix whose columns are the eigenvectors of the Hessian of $\mathcal{N}_H(\mathbf{x})$, and $\mathbf{T}(\mathbf{x}) : \mathbb{R}^n \to \mathbb{R}^{n \times n}$ is a diagonal matrix containing the corresponding eigenvalues. The condition $\nabla \wedge \mathbf{R}\nabla H = \mathbf{0}$ will be satisfied if we choose,

$$\mathbf{R}(\mathbf{x}) = \mathbf{Q}(\mathbf{x})\text{diag}\big(\mathcal{N}_\Lambda(\mathbf{x})\big)\mathbf{Q}(\mathbf{x})^{-1}, \tag{24}$$

where the operator $\text{diag}(\cdot)$ forms a square diagonal matrix from its vector argument, and $\mathcal{N}_\Lambda(\mathbf{x}) : \mathbb{R}^n \to \mathbb{R}^n$ gives the eigenvalues of $\mathbf{R}$. In this way, $\mathbf{R}$ and $\nabla^2 \mathcal{N}_H(\mathbf{x})$ share the same eigenvectors and so they will commute.

If we know that the system is energy decaying then we require the output of $\mathcal{N}_\Lambda(\mathbf{x})$ to be negative, which can be easily imposed by passing the outputs through the negative softplus function.

The further development of this parameterization is left as a direction for future work.

## C  Governing Equations

This section contains the governing equations for the numerical studies written in the form of a generalized Hamiltonian decomposition. Note that these decompositions are *not* unique. The Lorenz '63 decomposition was originally derived by Sarasola et al. [22] for the purpose of feedback synchronization of chaotic systems. Figure 8 shows some sample data used to train the models.

**Nonlinear Pendulum**

$$\mathbf{f}(\mathbf{x}) = \left( \begin{bmatrix} 0 & 1 \\ -1 & 0 \end{bmatrix} + \begin{bmatrix} 0 & 0 \\ 0 & -0.35 \end{bmatrix} \right) \begin{bmatrix} g\sin(x_1) \\ x_2 \end{bmatrix}. \tag{25}$$

**Duffing Oscillator**

$$\mathbf{f}(\mathbf{x}) = \left( \begin{bmatrix} 0 & 1 \\ -1 & 0 \end{bmatrix} + \begin{bmatrix} 0 & 0 \\ 0 & -0.35 \end{bmatrix} \right) \begin{bmatrix} x_1^3 - x_1 \\ x_2 \end{bmatrix}. \tag{26}$$

**Lorenz '63 System**

$$\mathbf{f}(\mathbf{x}) = \left( \begin{bmatrix} 0 & \sigma & 0 \\ -\sigma & 0 & -x_1 \\ 0 & x_1 & 0 \end{bmatrix} + \begin{bmatrix} \frac{\sigma^2}{\rho} & 0 & 0 \\ 0 & -1 & 0 \\ 0 & 0 & -\beta \end{bmatrix} \right) \begin{bmatrix} \frac{-\rho}{\sigma}x_1 \\ x_2 \\ x_3 \end{bmatrix}. \tag{27}$$

Figure 8: Pendulum, duffing oscillator, and Lorenz '63 sample data

# D    ODE Model Comparison Metrics

To compare different models, we will use two metrics: (i) the state error and (ii) the derivative error. To compute these metrics we first uniformly sample 50 initial conditions within the domain of interest (note the models have not seen these initial conditions) and simulate these initial conditions forward in time for 200 seconds using the true governing equations yielding,

$$\mathbf{X}_{true}^{(j)} = \{(\mathbf{x}_{true}(t_i), \dot{\mathbf{x}}_{true}(t_i)\}_{i=1}^{200} \quad \text{for} \quad j \in 1, 2, \dots, 50. \tag{28}$$

We then integrate these same initial conditions forward in time using the models,

$$\mathbf{X}_{pred}^{(j)} = \{\mathbf{x}_{pred}(t_i)\}_{i=1}^{200} \quad \text{for} \quad j \in 1, 2, \dots, 50, \tag{29}$$

and perform the following computations: (i) compute the mean $\ell^2$ norm of the difference between the predicted states and the true states,

$$\text{State error} = \frac{1}{50} \sum_{j=1}^{50} \frac{1}{200} \sum_{i=1}^{200} ||\mathbf{x}_{true}(t_i) - \mathbf{x}_{pred}(t_i)||_2, \tag{30}$$

and (ii) the mean $\ell^2$ norm of the difference between the predicted derivatives for each true state and the true derivatives,

$$\text{Derivative error} = \frac{1}{50} \sum_{j=1}^{50} \frac{1}{200} \sum_{i=1}^{200} ||\dot{\mathbf{x}}_{true}(t_i) - \text{model}(\mathbf{x}_{true}(t_i))||_2. \tag{31}$$

In all experiments, uncertainty estimates are given by one standard deviation from the mean.

# E    Neural Network Architectures and Hyperparameters

This section contains a complete description of the neural networks along with the hyperparameters used in this work. In all experiments, we used the Adam optimizer with a learning rate of $10^{-3}$ and a weight decay of $10^{-4}$.

**Section 4.1**    In this section we compare the weak form loss function to the state regression and derivative regression loss functions. In all cases, we use a FCNN with 3 hidden layers with 300 units in each hidden layer. All experiments use a batch size of 120. In the study which compared loss functions for learning ODEs, both the Weak form loss experiments and derivative regression experiments use 50 batch integration time steps while the state regression experiments use 10 batch integration time steps. This reduction in batch integration time steps was used in state regression experiments to cut the computational cost of the method to provide a realistic training time measurement for state regression. In our experiments state regression tended to require a lower batch integration time step than the other loss functions to achieve approximately the same performance. Appendix I below contains an experiment where the number of batch integration time steps was held constant for all methods.

**Section 4.2 – Nonlinear Pendulum**    In this experiment we used a GHNN with an asymptotically globally stabilizing prior, a HNN, and a FCNN. For all neural networks we selected 3 hidden layers with 300 units in each hidden layer.

All experiments used 100 batch integration time steps and a batch size of 120. Training data was generated by integrating two independent initial conditions forward for 20 seconds using the adaptive RK4(5) ODE integration tool provided by torchdiffeq [14] at a frequency of 50Hz. Independent zero mean Gaussian noise with a standard deviation of 0.1 was then added to these trajectories and used in training. For each experiment, 10 independent models were trained for each specific ODE parameterization. To choose between models, a validation dataset was created by integrating the initial conditions forward in time at 13Hz (meaning the validation dataset was 20% the size of the training set). Like for the training dataset, independent zero mean Gaussian noise with a standard deviation of 0.1 was added to these trajectories.

Testing data was generated by uniformly sampling 50 initial conditions (never before seen by the models) from within the domain of interest and integrating the trajectories forwards for 200seconds.

**Section 4.2 – Lorenz '63 System**   In this experiment we used a GHNN with a soft energy flux rate prior and a FCNN. The FCNN and GHNN were selected to have 3 hidden layers with 300 hidden units in each layer. Recall that for a GHNN with a soft energy flux rate prior we parameterize each component of the decomposition as follows: $H = \mathcal{N}_H$, $[\mathbf{J}]_{i,j} = \mathcal{N}_{i,j}$, and $\mathbf{R}\nabla H = \nabla \mathcal{N}_D$.

All experiments used 500 batch integration time steps and a batch size of 120. Training data was generated by integrating 21 independent initial conditions forward for 20 seconds using the adaptive RK4(5) ODE integration tool provided by torchdiffeq [14] at a frequency of 250Hz. Independent zero mean Gaussian noise with a standard deviation of 0.1 was then added to these trajectories and used in training. For each experiment, 10 independent models were trained for each specific ODE parameterization. To choose between models, a validation dataset was created by integrating the initial conditions forward in time at 63Hz (meaning the validation dataset was 20% the size of the training set). Like for the training dataset, independent zero mean Gaussian noise with a standard deviation of 0.1 was added to these trajectories.

Testing data was generated by uniformly sampling 50 initial conditions (never before seen by the models) from within the domain of interest and integrating the trajectories forwards for 200 seconds.

# F   $N$-Body Experiment Details

**$N$-body Forces**   We will consider $N$ particles in two dimensions. The particles all have unit mass and will impart the following gravitational forces on one another, with a gravitational constant of unity. The gravitational force felt on particle $i$ by a single particle $j$ is given by

$$F_{ij} = \frac{x^{(j)} - x^{(i)}}{\left\|x^{(j)} - x^{(i)}\right\|_2^3}, \tag{32}$$

where $x^{(i)} \in \mathbb{R}^2$ is the position of the $i$th particle. Summing over all particles yields the $N$-body equations of motion

$$\ddot{x}^{(i)} = \sum_{j=1, j \neq i}^{N} \frac{x^{(j)} - x^{(i)}}{\left\|x^{(j)} - x^{(i)}\right\|_2^3}, \qquad \text{for } i = 1, \dots, N. \tag{33}$$

The sum of potential and kinetic energy of the system is given by

$$H = -\sum_{1 \leq i < j \leq N} \frac{1}{\left\|x^{(j)} - x^{(i)}\right\|_2} + \sum_{i=1}^{n} \frac{\left\|v^{(i)}\right\|_2^2}{2}, \tag{34}$$

where $v^{(i)} \in \mathbb{R}^2$ is the velocity of the $i$th particle.

**Non-conservative Force Field**   The simulation will run in the presence of the non-conservative force field (which has a non-vanishing curl)

$$F = \text{sinc}\left(\sqrt{x_1^2 + x_2^2}\right)\left\{x_2, -x_1\right\}. \tag{35}$$

Note that this field is also divergence-free everywhere so that the vector field has no conservative component. The effect of this force field is that energy will be put into and taken out of the system throughout the simulation trajectory.

**Dynamics in the Force Field**   Combining the $N$-body dynamics in the force field gives the equations of motion

$$\ddot{x}^{(i)} = \sum_{j=1, j \neq i}^{N} \frac{x^{(j)} - x^{(i)}}{\left\|x^{(j)} - x^{(i)}\right\|_2^3} + \left\{x_2^{(i)} \text{sinc}(r^{(i)}), -x_1^{(i)} \text{sinc}(r^{(i)})\right\}^T, \qquad \text{for } i = 1, \dots, N, \tag{36}$$

where $r = \sqrt{x_1^2 + x_2^2}$ denotes the Euclidean distance of a particle from the origin. Writing this as a coupled system of first order ODEs gives the following $4N$ equations

$$\dot{v}^{(i)} = \sum_{j=1, j \neq i}^{N} \frac{x^{(j)} - x^{(i)}}{\left\|x^{(j)} - x^{(i)}\right\|_2^3} + \left\{x_2^{(i)} \text{sinc}(r^{(i)}), -x_1^{(i)} \text{sinc}(r^{(i)})\right\}^T, \qquad \text{for } i = 1, \dots, N, \tag{37}$$

$$\dot{x}^{(i)} = v^{(i)}, \qquad \text{for } i = 1, \dots, N. \tag{38}$$

Note that the potential energy of the system does not change from the $N$-body case since the force field is non-conservative, therefore the total energy relation is the same.

**Dataset Generation & Training Details** A dataset was generated by integrating the dynamics forward in time using $N = 12$ particles, giving $n = 4N = 48$ state variables. For initial conditions, all velocity state variables were initialized to zero, and all position state variables were sampled *iid* from a standard normal. The dynamics were integrated 30 units forward in time and 1500 state observations were taken, evenly spaced along the trajectory. Only a single trajectory was used for training.

Training was conducted using the using weak derivative regression with a 100 batch integration time steps and a batch size of 120.

## G Description of Data Mini-batching Hyperparameters

This section explains some intricacies around batching that are specific to learning ODEs. As was briefly mentioned in Section 3, our dataset can be written as follows: $\mathcal{D} = \left\{ \mathbf{x}_1^{(i)}, \mathbf{x}_2^{(i)}, \ldots, \mathbf{x}_T^{(i)} \right\}_{i=1}^{m}$ where we have used the notation $\mathbf{x}_j^{(i)}$ to indicate the measurement of the state at time instant $t_j$, $\mathbf{x}(t_j)$, for trajectory $i$. Recall that we collect $m$ trajectories of length $T$ of the state $\mathbf{x}$.

There are two batching hyperparameters that we have made use of in this work: (i) the batch size and (ii) the batch integration time steps. The batch size aligns with the typical notion of batch size while the batch integration time steps indicates the number of time steps ($l$) in each sub-sampled trajectory.

To be more clear, we sample a single training trajectory from our dataset as:

$$\mathbf{X}_{sample} = \left\{ \mathbf{x}_j^{(i)}, \mathbf{x}_{j+1}^{(i)}, \ldots, \mathbf{x}_{j+l}^{(i)} \right\}, \tag{39}$$

where $i$ and $j$ are random integers from the sets $\{1, 2, \ldots, m\}$ and $\{1, 2, \ldots, T - l\}$ respectively.

## H Description of Test Space

The test space was held constant throughout all experiments in this work. We used 200 Gaussian radial basis functions (GRBFs) which have been evenly spaced over the batch integration time window with a shape parameter of 10. For example, having sampled a batch integration time window given by $t_{sample} = \{t_j, t_{j+1}, \ldots, t_{j+l}\}$ where $j$ is a random integer from the set $\{1, 2, \ldots, T - l\}$, the test space is spanned by,

$$\text{GRBF}_k(t) = e^{-10(t-c_k)^2}, \quad \text{for } k = 1, 2, ..., 200, \tag{40}$$

where each $c_k$ is an evenly placed basis function center from the interval $[t_j, t_{j+l}]$.

## I Extended Study on Methods for Learning ODEs

This section contains an extended study on the performance weak form regression as compared to state regression and derivative regression. Here we have set the mini-batching hyperparameters to the same value for all loss functions to illustrate how the learning schemes perform given the same settings. Note that these hyperparameters were tuned for the experiment in Section 4.1 as it was observed that state regression could adequately recover the governing equations with a smaller number of batch integration time steps than the other methods. In this experiment, we trained a FCNN for 3000 epochs, with a batch size of 120, and a batch integration time of 50 steps. As before, we collect measurements of a nonlinear pendulum for 20 seconds as it decays towards its stable equilibrium from two independent initial conditions. We vary the measurement sampling frequency from 10Hz to 100Hz.

As above, the derivate regression loss function was unable to recover the the governing ODE at this level of noise. In particular, the loss function resulted in models whose trajectories diverged in finite time for all experiments but the 10Hz and 30Hz experiments; hence no state prediction errors were calculated for these models.

We see that the weak form loss function had a state error rate which improved as the measurement frequency was increased. This is expected given the fact that numerical integration schemes improve their performance as the spacing between quadrature points is decreased. We also see that state regression methods tended to lose accuracy as the measurement frequency was increased. This is expected given the fact that the the time interval over which integration is required is decreased as the measurement frequency is increased (ie. the ODE solver was required to integrate over a longer time window for each training step).

Most striking from this experiment is the fact that state regression required significantly longer training time than weak form regression. We observe that the weak form regression method had a training time which was constant with respect to the sampling frequency while state regression had a training time which increased with time between samples.

Figure 9: Comparison of State Error (L) and Derivative Error (R) for Different Loss Functions

Table 3: Loss Function Training Time Comparison

| Method | \multicolumn{10}{c}{Measurement Frequency (Hz)} | | | | | | | | | |
|---|---|---|---|---|---|---|---|---|---|---|
| | 10 | 20 | 30 | 40 | 50 | 60 | 70 | 80 | 90 | 100 |
| Weak form | 0:00:32 | 0:00:32 | 0:00:32 | 0:00:32 | 0:00:32 | 0:00:32 | 0:00:33 | 0:00:31 | 0:00:31 | 0:00:32 |
| State regression | 4:23:59 | 3:12:15 | 2:26:59 | 2:20:32 | 2:20:37 | 2:10:38 | 2:08:58 | 2:09:52 | 1:56:48 | 1:45:22 |
| Derivative regression | 0:00:28 | 0:00:29 | 0:00:28 | 0:00:28 | 0:00:29 | 0:00:27 | 0:00:29 | 0:00:29 | 0:00:30 | 0:00:31 |

## J  Duffing Oscillator Experiment

In this section we demonstrate GHNNs applied to learning a generalized Hamiltonian decomposition of the Duffing oscillator. A generalized Hamiltonian decomposition of the governing equations for the Duffing oscillator is provided in Appendix C. We measure the system at a frequency of 50Hz as it decays towards stability using 10 independent initial conditions. Measurements are corrupted by zero mean Gaussian noise with a standard deviation of 0.1.

In this experiment we place a prior on the form of the governing equations which enforces local stability. Recall that this prior ensures that energy must strictly decrease along trajectories even for a set of randomly initialized neural network weights. The learnt generalized Hamiltonian and vector field is shown in Figure 11.

Again, we observe that our model learnt the important qualitative features of the vector field, $\mathbf{f}$, and generalized Hamiltonian. Furthermore in Figure 10 we observe that the trajectories produced by the learnt model align well with trajectories produced by the true underlying equations and the generalized Hamiltonian energy function. The performance of GHNNs on this problem is compared to FCNNs and HNNs in Table 2.

Figure 10: GHNN predicted trajectory example

For all neural networks we selected 3 hidden layers with 300 units in each layer. For each experiment, 10 independent models were trained for each specific ODE parameterization. To choose between models, a validation dataset was created by integrating the training initial conditions forward in time at 13Hz (meaning the validation dataset was 20% the size of the training set). Like for the training dataset, independent zero mean Gaussian noise with a standard deviation of 0.1 was added to these trajectories.

Testing data was generated by uniformly sampling 50 initial conditions (never before seen by the models) from within the domain of interest and integrating the trajectories forwards for 200 seconds.

Figure 11: Learnt (L) and true (R) generalized Hamiltonian and vector field for the Duffing oscillator

## K   Lorenz '63 Experiment Without Prior

In this section we demonstrate GHHNs applied to learning a generalized Hamiltonian decomposition of the Lorenz '63 system when no prior is placed onto the form of the generalized Hamiltonian. We use the same data as we did for the experiment in Section 4.2. In Figure 12, we observe that without the soft energy flux rate prior, we are still able to learn a reasonable approximation to the underlying governing equations. To reiterate the discussion above, we expect to only be able to capture qualitative aspects of the trajectory due to the fact that the Lorenz '63 equations are chaotic. As we would expect, without placing a prior onto the form of the generalized Hamiltonian, we see in Figure 13 that we are unable to recover the specific generalized Hamiltonian decomposition given in Appendix C. As the general-

Figure 12: Lorenz predicted trajectory example with no prior

ized Hamiltonian decomposition is not unique, we expect to learn a generalized Hamiltonian which does not necessarily align with the arbitrary decomposition chosen in Appendix C. This example draws attention to the fact that there is a potential for future work in reducing the space of plausible generalized Hamiltonians for general dynamical systems.

Figure 13: Lorenz generalized Hamiltonian and vector field: no prior (L), soft prior (C), and true (R)