[Reviews · NeurIPS 2020]

Review 1

Summary and Contributions: This paper proposes a method to learn a generalized Hamiltonian decomposition (learning an energy function). This method applies to more general scenarios than previous work.

Strengths: Since this method applies to more general problems than previous work, it might inspire more research on this more general direction. They test it on noisy & chaotic systems. They are able to incorporate physical priors via special parametrizations of neural networks. I appreciate the clear distinction between "weak form regression," "strong regression," and "state regression." I have done some work with the latter two, but no the former. As mentioned by the authors, this may be the first paper to approach the problem as "weak form regression" in a deep learning context.

Weaknesses: As I'll detail below, I found the paper hard to follow. I think it would be difficult to reproduce without further clarifying details. I also found the experiments unconvincing.

Correctness: There aren't any explicit references to held out (test) data. Is that what's meant in Appendix D (ODE Model Comparison Metrics)? Are those 50 initial conditions different than the ones used for training? If so, are all reported metrics & figures about results from these 50 held-out initial conditions? My understanding is that Section 4.1 compares ways to learn a pendulum model with a fully connected neural network that is not concerned with learning an energy function. This section focuses on comparing three approaches to learning the dynamics model: weak form regression, derivative regression, and state regression. We see that weak form regression has significantly lower error than the other two options. However, I was surprised that all of the errors were so high. Similarly, the later sections, which add in the aspect of learning the generalized Hamiltonian, have significant error. Are there any papers we can compare to with similar examples at a similar level of noise? See, for example Rudy, et al. "Deep learning of dynamics and signal-noise decomposition with time-stepping constraints" JCP, 2019 for an example of seemingly much more accurate results to similar problems. Also, looking at Figure 2 in [17], their state error seems lower on their real pendulum than your state error on the noisy pendulum. However, perhaps your problems are much harder in a way that is not obvious to me. I think it would help to report relative error & plot what the noisy data looks like, as is done in the Rudy, et al. paper. If the baselines in this paper are not very representative of how accurate other methods are, then that greatly weakens the claim that your method is an improvement. I added one more "correctness" comment under "additional feedback" since I ran out of space in this box.

Clarity: In general, I found this paper hard to read. However, I realize that I may be missing some important background pieces. Here are some suggestions: I found it hard to follow where the neural network comes in for the weak form learning (Equation 16). Since inference for an input convex network is itself a convex optimization problem, how does using input concave neural networks for parameterizations affect the speed of inference? (I know I asked about training time already, but is inference time also important to consider here?) In Appendix D, explaining the metrics, it says "We then integrate these same initial conditions forward in time using the models and perform the following computations..." Are the training losses the log likelihoods in Section 3, but then to compare the models, you use this proceduce described in Appendix D? When you talk about integrating forward using the models, does that mean using an ODE integrator? i.e. for derivative regression, were the derivatives from the neural network then plugged into an ODE integrator? Was it the same integrator as described in Section 4 that you used for state regression? Similarly, how were the derivatives estimated for the sake of these metrics? (other than for the derivative regression problem.) Could you clarify how to check if you have learned an energy function? I know for a Hamiltonian, I would check if it is conserved along trajectories, but for an unknown generalized Hamiltonian, I'm not sure how to check if I learned one well. I'm confused about the "setting energy flux rate" prior. This method is used to learn a scalar energy function. If you don't already know the energy function, then how do you know the energy flux rate?

Relation to Prior Work: I found it hard to tell in Section 2.1 what was required to extend the generalized Hamiltonian decomposition in [22] from R^3 to R^n. Is the main change replacing the Helmholtz decomposition with the Helmholtz Hodge decomposition?

Reproducibility: No

Additional Feedback: Last correctness comment: In Section 4.2, it's stated that "In all cases, GHNNs perform at least as well as the other state-of-the-art continuous time models while simultaneously learning, generalized Hamiltonian energy function and the energy cycle for the system." State-of-the-art is too strong of a strong claim. There are decades of research on using neural networks to predict dynamical systems. You compare against two models: a fully-connected neural network that may or may not have been tuned, and a Hamiltonian Neural Network which was actually designed to learn Hamiltonians. (If you train a HNN on data that doesn't have a Hamiltonian to conserve, are you putting it at a disadvantage?) Worse, your model is only the most accurate of the 3 occasionally in Table 2, so "at least as well as" is not correct. UPDATE: Thank you for addressing several of my concerns in the response. I updated my score from a 4 to a 5.


Review 2

Summary and Contributions: The paper proposes two significant contributions (generalised hamiltonians + weak-form optimisation), and either one would have been alone sufficient for publication. The paper presents novel concepts and pushes the boundary in neural ODEs.

Strengths: The work is sound, significant, very novel and relevant.

Weaknesses: The experiments are limited, the connection to realistic usecases / applications could have been stronger

Correctness: Everything looks correct.

Clarity: Clearly written. The material is technical but the paper does a good job presenting the concepts. Some additional cartoon visualisations would have still helped.

Relation to Prior Work: Very well described and positioned

Reproducibility: Yes

Additional Feedback: The paper proposes hamiltonian ODE systems via helmholtz-type decompositions into curl and div-based fields. They propose neural network parameterisation to learn such systems, and propose a range of priors and model choices to more realistically model and represent ODE systems. Second, they propose new weak-form optimisation technique that seems very promising. Both contributions are very significant and will certainly lead to high impact in deep learning / diffeqs. The paper is excellently written, I enjoyed reading it a lot. The main drawback of the paper is limited experiments. The weak-form optimisation is an excellent contribution, but its superiority has not yet been convincingly demonstrated based on these very limited experiments. Likewise, table 2 shows that the new model is not dramatically better than earlier approaches. The paper would have improved considerably with realistic applications or benchmarks instead of the toy’ish systems. Despite these drawbacks, this is a fantastic paper that I’m sure will interest the nips community, and I’d be happy to see it published. Minor comments o The theorem 1 seems incomplete. The theorem is true for any skew-symmetric J’s, but the connection to g() is vague. With arbitrary g the J is not guaranteed to be skew-symmetric. Parameterising only one triangle of J with g should work. The proof is correct. o Its unclear if eq 7 means that all g_ij’s are same network, or if they are different networks.


Review 3

Summary and Contributions: The authors provide a new method for learning an ODE that describes time series data, offering both a new (potentially more interpretable) technique for parametrizing the fitted ODE and a new technique for training the parameters.

Strengths: This is a very well-written paper. The problem that it wants to solve is quite clear, and the descrptions of the two methodological advances are easy to follow.

Weaknesses: The numerical experiments are convincing, but it would have been nice (if the authors had more room) to include an interpretation of the learned ODE parameters and to include a more real-world, not as simulated example. This might have illustrated some limitations of the method, as when some data is hidden, learning a first order ODE may perform badly. Ultimately, the failure of the authors to provide a real-world example led to me docking a point in the final review.

Correctness: As far as I can tell, the claims and the methods are correct.

Clarity: Yes.

Relation to Prior Work: Yes; this is particularly well-done.

Reproducibility: Yes

Additional Feedback:


Review 4

Summary and Contributions: The authors discuss the generalized Hamiltonian decomposition of ODEs and demonstrate its use in estimating the vector field in ODEs.

Strengths: The generalized Hamiltonian decomposition offers an physically intuitive way to describe the dynamics.

Weaknesses: As it pertains to parameter estimation some things are not explained well. Additionally, some aspects of the proposed weak from approach are not discussed in sufficient detail. I elaborate below.

Correctness: Appears so.

Clarity: So-So.

Relation to Prior Work: Yes.

Reproducibility: Yes

Additional Feedback: I considered the author reply, however I remain convinced that the proposed weak from regression would require some closer examination. 1) I guess strictly speaking, it should be \circ Identity map(x) rather than \circ x in the definition of N(x). 4) It seems that one drawback of the weak form in Eq. (15) is that, while yes quadrature can be used, you are restricted to low order quadratures since you can not evaluate x(t) at any t? Also how sensitive is this approach to noise in comparison to other methods? 5) The paranthesis in Eq. (16) around the argument of p could be made bigger

[Author Response · NeurIPS 2020]

First, we would like to thank the reviewers for their thoughtful critiques and commendations. We appreciate and are
energized by the fact that reviewers found our work to be significant and novel (R2) while positioning itself well with
respect to previous work (R2, R3). We have done our best to respond to as many questions and concerns as possible:

**While [R3] found the experiments convincing, reviewers also noted that the paper would have been more im-**
**pactful by including a broader range of experiments that are more realistic / application focussed (R1, R2, R3).**
We agree the paper would certainly have been strengthened with real-world applications. The intent behind focusing
on model problems was to examine the fundamental properties of GHNNs given basic physical systems – leaving
applications on complex real-world systems to future work.

**[R1] asked about whether metrics were produced using 50 initial conditions from a held-out test set.** This was
indeed the case and we have made it more clear in our revised paper. Further to this, **[R1] raised some questions**
**about the clarity of the metrics in Appendix D**. We agree we could have done a better job explaining our comparison
metrics. For this reason we now have added equations describing the metrics in detail to clear up these concerns.

We want to be clear that we do not wish to claim that **weak regression performs significantly better than state**
**regression (R1).** Rather we claim that "[in our example] weak derivative matching has comparable performance to
state regression while requiring substantially less runtime." **[R1] also notes that our claim that "GHNNS perform at**
**least as well as other state-of-the art continuous time models"** is too strong. This is a fair comment and our choice
of wording could have been better. We have remedied this claim and have added an extended discussion explaining
why we believe GHNNs perform approximately as well as the other modern models for continuous time ODEs while
simultaneously learning an underlying energy function.

Thank you **[R1] for sharing an excellent paper from Rudy et. al. [R1] asked about why they appeared to have**
**a lower error on similar problems**. We would like to point out that, while they do a form of state regression in
their paper, they use a slightly different metric to compute the performance of their models. More specifically: (i)
they compute errors based on the single initial condition they have access to at training time and (ii) they compute a
*normalized* error metric that they call the $E_F$ score. We compute a mean $E_F$ score of approximately 0.06 for the weak
form regression method and a mean $E_F$ score of 0.27 for the state regression method at a noise of approximately 20% –
aligning with their $E_F$ score of approximately $0.23 \pm 0.3$ for their cubic oscillator (note that we haven't done a full
study to validate this $E_F$ score – we only made use of the two training trajectories and the pretrained models we had
used to produce Table 1). Thank you **[R1] for suggesting we plot the noisy data on top of a nominal trajectory as**
**was done in this paper**. We agree this would make our exposition more clear and we have added these plots to the
revised paper.

**[R1] also asked about why Figure 2 in [17] seemed to show lower state errors on a real pendulum than our**
**simulated pendulum**. We note that the real pendulum data is significantly less noisy than our simulated data (see
Schmidt et. al. "Distilling Free-Form Natural Laws from Experimental Data", Science, 2009). Furthermore, they only
model the system for approximately 20 seconds over which time their system can be approximated as energy conserving
– making it a good candidate for HNNs.

**[R2] points out that theorem 1 is unclear**. Thanks for pointing this out, we've updated the proof to make it clear
that $g_{ij} = g_{ji}$ in our parameterization. **[R2] also notes that it's not clear if all $g_{ij}$'s are the same network**. We've
updated our paper so that eq. (7) now reads $g_{i,j}(\mathbf{x}_{\setminus ij}) = \mathcal{N}_{i,j}(\mathbf{x}_{\setminus ij})$ to make it clear that these are different networks.

**[R4] asked if the method is still applicable in the case m=1.** We can set m=1 without changing the method presented
here. The number of independent trajectories required will depend on the complexity of the ODE. We noted the number
of independent trajectories used for each experiment in the original submission.

**[R4] asks if the measurements are assumed to be exact**. We assume various amounts of zero mean Gaussian noise
on our measurements that are listed in the original submission. We believe our experiments show our methods are
effective given noisy measurements.

**[R4] also notes that a draw back of weak form regression is that you are restricted to low order quadratures**
**since you cannot evaluate x(t)**. This is true in the way we have presented our work here. We briefly discuss this
in Appendix H where we show how state regression outperforms weak derivative regression when the measurement
sampling frequency is low. That being said, there exists a suite of methods from the state estimation / data assimilation
fields for estimating $\mathbf{x}(t)$ given an uncertain measurement and dynamics model. We have mentioned this possible
extension in our newly added "future work" section.

[Meta-Review · NeurIPS 2020]

There was significant discussion of this paper post rebuttal. I am following R2's argument that the core idea of this paper is not just very strong, but also very timely. The main issues raised by the other reviewers are a) that the paper is technically dense. It is debatable whether much can be done about this. The paper arguably does a good job of introducing the heavy algebraic geometry machinery, although one can wonder whether there may not be a simpler way of presenting the same ideas. b) the experiments are underwhelming. They are technically well done, but the setups are not ideal. Especially the second example, the Lorentz attractor is perhaps not a good choice. Its chaotic nature is irrelevant to the points of the paper. A higher-dimensional, more motivating example (e.g. many-body dynamics, or kinematics of a humanoid robot, etc.) would be a better fit for the NeurIPS community. Nevertheless, the contributions of this paper are so valuable, and the development in this domain currently so rapid, that I believe there should be a place for this paper in *this year's* NeurIPS. I encourage the authors to take on the points raised by the reviewers and improve the paper in time for camera ready.